# Using electronic medical records to understand the impact of SARS-CoV-2 lockdown measures on maternal and neonatal outcomes in Kampala, Uganda

Joseph Ouma[1], Lauren Hookham[2]*, Lorna Aol Akera[1], Gordon Rukundo[1], Mary Kyohere[1], Ayoub Kakande[3], Racheal Nakyesige[3], Philippa Musoke[1], Kirsty Le Doare[1,2]

**1** Makerere University Johns Hopkins University Research Collaboration, Kampala, Uganda, **2** St. George's University of London, London, United Kingdom, **3** Medical Research Council/Uganda Virus Research Institute and London School of Hygiene and Tropical Medicine Uganda Research Unit, Entebbe, Uganda

* lhookham@sgul.ac.uk

**Data Availability Statement:** Data are available in a public, open access repository (Figshare) at St

## Abstract

Kawempe National Referral Hospital (KNRH) is a tertiary facility with over 21,000 pregnant or postpartum women admitted annually. The hospital, located in Kampala, Uganda, uses an Electronic Medical Records (EMR) system to capture patient data. Used since 2017, this readily available electronic health record (EHR) has the benefit of informing real-time clinical care, especially during pandemics such as COVID-19. We investigated the use of EHR to assess risk factors for adverse pregnancy and infant outcomes that can be incorporated into a data visualization dashboard for real time decision making during pandemics. This study analysed data from the UgandaEMR collected at pre-, during- and post-lockdown timepoints of the COVID-19 pandemic to determine its use in monitoring risk factors for adverse pregnancy and neonatal outcomes. Logistic regression models were used to identify the risk factors for adverse pregnancy and maternal outcomes including prematurity, obstetric complications, still births and neonatal deaths. Pearson chi-square test was used for pairwise comparison of the outcomes at the various stages of the pandemic. Data analysis was performed in R, within the International COVID-19 Data Alliance (ICODA) workbench. A visualisation dashboard was developed based on the risk factors, to support decision making and improved healthcare delivery. Comparison of pre-and post-lockdown variables showed an increased risk of pre-term birth (adjusted Odds Ratio (aOR = 1.67, 95% confidence interval (CI) 1.38–2.01)); obstetric complications (aOR = 2.77, 95% CI: 2.53–3.03); immediate neonatal death (aOR = 3.89, 95% CI 2.65–5.72) and Caesarean section (aOR = 1.22, 95% CI 1.11–1.34). The significant risk factors for adverse outcomes were younger maternal age and gestational age <32weeks at labour. This study demonstrates the feasibility of using EHR to identify and monitor at-risk subpopulation groups accessing health services in real time. This information is critical for the development of timely and appropriate interventions in outbreaks and pandemic situations.

George's University, London. The link to the dataset is DOI: 10.24376/rd.sgul.22116965.

**Funding:** This study was funded through the Grand Challenges ICODA pilot initiative, delivered by Health Data Research UK by the Bill & Melinda Gates Foundation and the Minderoo Foundation to KLD. Research reported in this publication was also supported by the National Institutes of Health's Fogarty International Center's Training award number D43TW012275 to KLD. The content is solely the responsibility of the authors and does not necessarily represent the official views of the National Institutes of Health or the Health Data Research UK. The funders had no role in study design, data collection and analysis, decision to publish, or preparation of the manuscript.

**Competing interests:** The authors have declared that no competing interests exist.

**Abbreviations:** ICODA, International COVID-19 Data Alliance; IROC, Incidence and Risk Factors for COVID-19 amongst Pregnant and Lactating Women and their Infants in Uganda; KNRH, Kawempe National Referral Hospital.

## Introduction

The direct effects of COVID-19 can be measured by the number of cases and associated morbidity and mortality data. However, changes in health service delivery or utilization secondary to lockdown restrictions will also impact upon the health and wellbeing of the population indirectly. Data from early in the pandemic showed the negative impact of lockdown restrictions on maternal and neonatal health, with an increase seen in pregnancy complications such as stillbirth and preterm birth, increase in admissions to neonatal intensive care units and neonatal deaths, as well as decreased frequency of routine vaccination for infants [1]. Availability of accurate, timely and reliable data for monitoring the risk factors for these negative outcomes is therefore critical for improving the safety and well-being of both mother and child, especially during periods of health care disruption caused by pandemics.

Electronic Health Records (EHRs) are becoming widely available in Uganda and other low resource settings and have reduced the cost of capturing and storing patient data compared to traditional paper-based tools; improved tracking of patients and clinic efficiency secondary to availability of patient data [2]. When combined with dashboards to present summary data, including user defined interfaces and reports, such large datasets can be used by healthcare workers and facility managers for planning and monitoring purposes [3]. Uganda first piloted and adopted an open-source electronic medical records system, the UgandaEMR, in 2011, now in use in more than 1000 health facilities across the country [4]. The system enables unique identification and tracking of patients throughout their hospital attendances and reporting & sharing of data across departments (e.g., Pharmacy, Laboratory, Clinic). The system is also linked to the national District Health Information System (DHIS2) and therefore facilitates direct reporting on key health services performance indicators. Development partners such as UNAIDS, WHO and UNICEF have supported use of EHRs to monitor service delivery and inform improvement in healthcare delivery decisions in Uganda [5–7]. The Ministry of Health and these development partners have also invested in improving the quality of data from these systems through monthly and quarterly monitoring and reviews, capacity building and strengthening of infrastructures to support electronic data capture and management at health facility level [2].

Our primary hypothesis was that EHR could be used to identify risk factors for adverse pregnancy and infant outcomes during pandemics such as COVID-19 to inform real-time clinical care decisions. We aimed to investigate changes in risk factors in three discrete time periods: pre-, during- and post-pandemic lockdown using routine EHR. We also sought to incorporate these risk factors for adverse pregnancy and infant outcomes in a user-friendly dashboard interface that could be utilised by both the local health facility and Ministry of Health to monitor and predict adverse health outcomes and introduce risk mitigation measures in real time during this and future outbreaks and pandemics.

## Methods

### Data and setting

Kawempe National Referral Hospital (KNRH) is a tertiary women and children's hospital based in Kampala, Uganda. It is a high-volume health facility with over 21,000 deliveries, 40,000 antenatal attendances, 45,000 PMTCT attendances and 28,000 BCG vaccination clinic visits. The Hospital uses UgandaEMR to capture information on all individuals (men, women and children) accessing healthcare services at the facility. The UgandaEMR is installed on a central server at Kawempe National Referral Hospital and accessible through a web browser. The system captures data for antenatal, delivery and postnatal attendance at the hospital.

Health care workers write patient information on paper tools, based national Health Management Information System (HMIS), supplied by Ministry of Health, Uganda. The information is then transcribed into the electronic information system by records clerks. The system has inbuilt data quality and integrity checks, as well as customized reports for Ministry of Health reporting. Each user has a unique username and password. The development and maintenance of the system is supported by the Makerere University Monitoring and Evaluation Technical Support project (METS) funded by President's Emergency Plan for AIDS Relief (PEPFAR)-USA through the Centers for Diseases Control (CDC), Uganda. The research techniques adhere to the STROBE (Strengthening the Reporting of Observational Studies in Epidemiology) Statement [8].

We analyzed data from the hospital antenatal, maternal and neonatal attendance for the period January 2020 to October 2021. The variables included in the analysis were maternal age (defined as <20, 20–24, 25–34 or 35+ years), gravidity (primigravida, multi-gravida (<5 pregnancies)-low or multigravida (> = 5 pregnancies -High), parity (nullipara, primipara or multipara), obstetric diagnosis—the clinician evaluation of the mode of delivery (born before arrival, caesarian section or spontaneous vaginal delivery), obstetric complications defined as a mother experiencing any complications before or at the time of admission to labour or in the peripartum period requiring an intervention -yes or no; these complications included e.g. pre-eclampsia, antenatal or postnatal hemorrhage and malaria in pregnancy etc. and these were all combined as composite variable), gestational age at time of hospital admission or at onset of labour (<28, 28–32 or 35+ weeks) and maternal status at discharge (alive or dead). Infant data included birth weight (<1.5, 1.5–2.4 or 2.5+ Kilograms), APGAR score at 1 and 5 minutes (0, 1–6 or 7+), condition of infant at birth (live birth, Immediate neonatal death or still birth) and whether the infant was born prematurely/delivery outcome (Preterm and term).

The study used cross-sectional panel data collected and categorized as Pre-lockdown (1st January–17 March 2020), Lock down (18 March 2020 to 25th May 2020 and 7th June 2021 to 31st July 2021) and Post Lockdown: 26th May 2020 to 6th June 2020 and 1st August 2021 to 31st Oct 2021). During these periods, the country experienced varying levels of Government restrictions/containment measures to avert the spread of COVID-19 (S1 Table).

## Statistical analysis

Descriptive statistics on maternal characteristics, and outcome of mother and infant during antenatal attendance, admission to labor and at discharge were summarized and reported. Means and standard deviations were reported for continuous variables and where the data was not normally distributed, medians and interquartile range were reported. Frequencies and percentages were computed and reported for categorical variables. Chi square and median tests were used for the pairwise comparison of outcomes for the pre- versus during and post COVID-19 lockdown time points for the categorical and continuous variables respectively. Results are presented, comparing outcomes for the period pre-lockdown versus lockdown and the period pre-lockdown versus the post lockdown period separately. We also present, graphically antenatal and delivery attendance volumes over the period of analysis.

Multinomial regression models were used to calculate prevalence (odds) ratios for adverse pregnancy and maternal outcomes for the period before COVID-19 lockdown (pre-lockdown), during the COVID-19 lock down (lockdown) and immediately post-lockdown period. Binary logistic regression model was used to assess the risk factors for adverse delivery outcome (pre-term births) and whether mother had an obstetric complication while multinomial logistic regression was applied to ascertain the risk factors for still birth, immediate neonatal

death, likelihood of caesarean section and for children born before arrival to hospital. All the models were adjusted for COVID-19 lockdown phases (pre-, during and post- lockdown). Crude and Adjusted Odds ratios were computed after controlling for age, parity, gravidity and gestational age. Data analysis was carried out in R, within the ICODA workbench (Aridhia) [9]. The workbench provided pre-developed and well tested statistical tools and models for statistical analysis and presentation of the analysis results.

Data used for this study is routine service delivery data and therefore had a high level of missingness. Variables considered for the analysis were those with less than 30% missingness overall, irrespective of the period this occurred. S2 Table presents the level of missingness and the statistical models applied to impute for missing data for each variable considered in the analysis. Multiple Imputation via Chained Equations (MICE) was used to impute missing values, because it allows for application of specific and suitable models based on the distribution of the observed values of the variable (outcome) of interest.

## Methodology—Dashboard development

To comprehensively develop the dashboard, we solicited stakeholder input through consultative meetings with officials from Ministry of Health and KNRH. During the stakeholder meetings, data on the level of completeness, consistency and validity were shared and strategies to improve the quality of data collected and entered into the system were sought. Further consultations in the form of one-to-one meetings, presentations and chart review were conducted to refine the strategies developed and the outlay of the dashboard. Microsoft Power BI Desktop version 2.104.702.0, and the UgandaEMR system were integrated using SQL connector. Anonymised data from the different tables in the UgandaEMR database were extracted using SQL queries. Data analysis expressions were then written in Power BI to collate data relating to the risk factors identified during risk factor modelling and summaries are presented on the dashboard with dynamic filters for days, weeks, months, and years as may be desired by users. Fig 1 presents a sample of the dashboard interface.

## Ethics statement

Ethical clearance to conduct this study was obtained from the Makerere University School of Medicine Research Ethics Committee SOMREC (#2020–089), Uganda National Council of Science and Technology (#HS623S) and St George's School of Medicine Research Ethics Committee (#2020.0146). All individual-level data was identified using unique person identifiers only, names and other personally identifiable information were excluded.

## Consent

Data were extracted from an anonymous Ministry of Health data records system. Healthcare providers seek verbal consent from all women to before recording their information on the paper forms and thereafter provide the needed healthcare (antenatal, delivery and postnatal care services). For neonates, written informed consent is provided by parent(s)/guardian. If a mother died during the course of health care access and the infant's father had not given consent, then the infant's guardian was asked to provide informed consent for the continued provision of care to the baby. Information about services available in the hospital and possible choices is provided during a general health talk to all women and their caregivers every morning before service delivery commence.

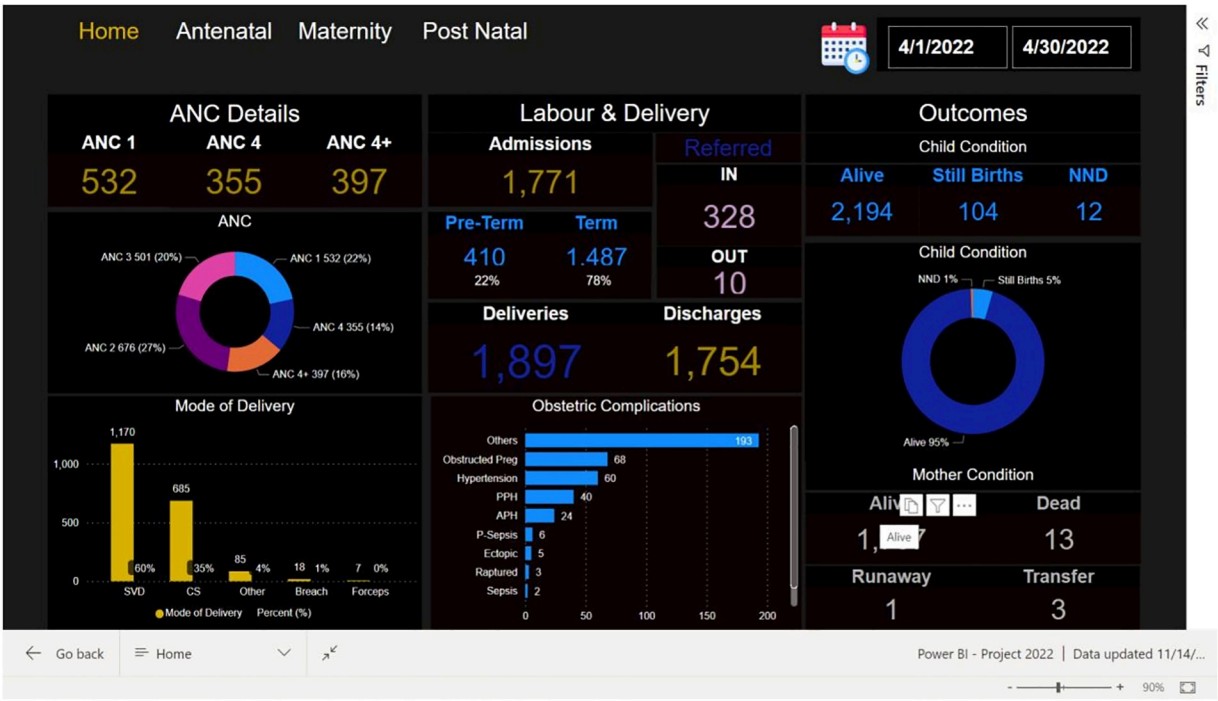

**Fig 1. Sample dashboard layout showing ANC and labour attendance and outcomes for both mother and infant.** A summary dashboard page showing antenatal attendance, labour and delivery outcomes for the period 1st April 2022 to 30th April 2022.

## Results

Between January 1st 2020 and October 31st, 2021 there were 15,870 first ANC attendances, 41,287 deliveries, and 42,870 post-natal attendances. Over 45,000 women were tested for HIV. The median number of monthly deliveries was 1,905 (IQR: 1,823–1,976). Figs 2 and 3 present

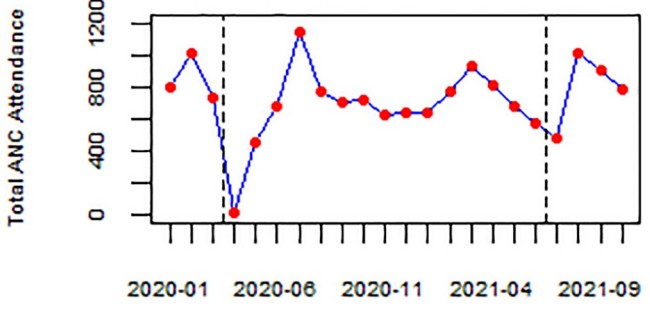

**Fig 2. Total antenatal attendance: Jan 2020 to Oct 2021.** Fig 2 shows monthly total ANC attendance for the period January 2020 to October 2021 at KNRH. Vertical dotted lines indicate timepoints (months) when total lockdown was in place.

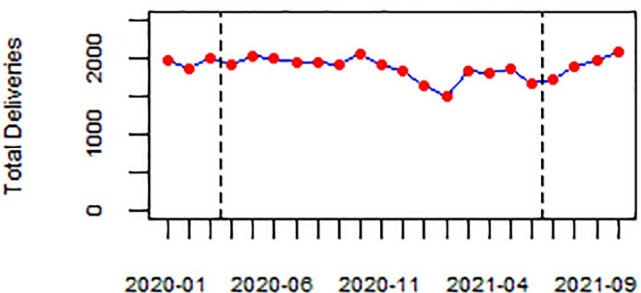

**Fig 3. Total number of deliveries: Jan 2020 to Oct 2021.** Fig 3 shows monthly total deliveries for the period January 2020 to October 2021 at KNRH. Vertical dotted lines indicate timepoints (months) when total lockdown was in place.

the monthly total antenatal attendances and deliveries in the hospital for the period January 2020 to October 2021. Figs 2 and 3 presents service volumes over the period of the study.

## Maternal and infant characteristics

Most of the women were aged 25-34years (2,514 (43.1%) pre-, 3,824 (44.9%) during- and 5,904 (43.6%) post-lockdown; most of the women were multipara (2,081 (35.7%) pre-, 3,081 (36.2%) during- and 5,154 (37.0%) post-lockdown). Maternal and infant characteristics can be seen in Table 1.

## Impact of COIVD-19 lockdown and risk factors for adverse pregnancy and infant outcomes

Maternal and neonatal outcomes at the different time points (pre-lock down, lockdown and post-lockdown) are presented in Table 2 and the risk factors for adverse pregnancy outcomes are presented in Table 3.

**Prematurity.**   The odds of delivering a premature infant increased during the lockdown and post-lockdown periods compared to the period before the lockdown (adjusted Odds Ratio (aOR) = 1.35(95% confidence interval (CI): 1.18–1.56) and (aOR = 1.67 (95% CI: 1.38–2.01) respectively (Table 2).

The odds of preterm birth were highest in younger mothers (age<20, aOR = 1.26: 1.14–1.40) compared to mothers aged 25–34 years. Odds of pre-term births were higher in the age group 35+ years compared to those aged 25-34years, although not statistically significant. Other risk factors associated with prematurity included having >5 previous pregnancies (aOR 1.27, 1.05–1.55) compared to Primigravida. The odds of prematurity were highest in the post-lockdown period (aOR 1.66, 1.38–2.00) compared to the period before the outbreak of COVID-19 (Table 3).

**Obstetric complications.**   Maternal complications increased during and after lockdown. The odds of maternal complication were highest in periods of post-lockdown (aOR = 2.77, 95% CI: 2.53–3.03) compared to the period pre-lockdown. The likelihood of Obstetric

**Table 1. Maternal and baby characteristics and outcomes.**

| Characteristic | Pre-Lockdown (n = 5,833) | Lockdown (n = 8,519) | Post-Lockdown (n = 13,544) | P1 | P2 |
|---|---|---|---|---|---|
| **Maternal Age (Years)** | | | | | |
| <20 | 773 (13.3) | 984 (11.6) | 1,661 (12.3) | 0.013 | 0.118 |
| 20–24 | 2,050 (35.2) | 2,984 (35.0) | 4,730 (34.9) | | |
| 25–34 | 2,514 (43.1) | 3,824 (44.9) | 5,904 (43.6) | | |
| 35+ | 494 (8.5) | 726 (8.5) | 1,246 (9.2) | | |
| (Missing = 6) | | | | | |
| **Parity** | | | | | |
| Nullipara | 1,855 (34.5) | 2,640 (34.2) | 4,193 (33.2) | 0.380 | 0.039 |
| Primipara | 1,435 (26.7) | 2,003 (25.9) | 3,301 (26.1) | | |
| Multipara | 2,081 (38.8) | 3,081 (39.0) | 5,154 (40.8) | | |
| (Missing = 2152) | | | | | |
| **Gravidity** | | | | | |
| Primigravida | 1,725 (33.9) | 2,524 (33.9) | 4,088 (34.0) | 0.809 | 0.857 |
| Multigravida-Low | 2,685 (52.8) | 3,908 (52.4) | 6,296 (52.4) | | |
| Multigravida-High | 678 (13.3) | 1,020 (13.7) | 1,638 (13.6) | | |
| (Missing = 3331) | | | | | |
| **Gestational age (Weeks),** | | | | | |
| Pre-term (<28 weeks) | 71 (1.4) | 93 (1.3) | 156 (1.5) | 0.195 | 0.127 |
| Pre-term (28–32 weeks) | 92 (1.9) | 166 (2.3) | 255 (2.4) | | |
| Term (>32 weeks) | 4,795 (96.7) | 6,938 (96.4) | 10,365 (96.2) | | |
| (Missing = 2,197) | | | | | |

P1 -P value for Chi-square test comparison between Pre- Lockdown and Lockdown time period, P2- P value for Chi-square test comparison between Pre-Lockdown and Post Lockdown time period

complication was also higher during the lockdown period compared to the period before the lockdown (aOR = 2.62, 95% CI 2.38–2.88) (Table 2).

The odds of obstetric complications were highest among women who gave birth at gestation <28 weeks and 28-32weeks compared to those who gave birth at gestation >32weeks (aOR = 1.42: 95% CI 1.19–1.69; and aOR = 1.38, 95% CI: 1.26–1.51 respectively) (Table 3).

**Still-births and immediate neonatal deaths.** Whilst there was no statistically significant increase in the odds of stillbirth during lockdown and post-lockdown compared to pre-lockdown period, the odds of immediate neonatal deaths increased significantly during lockdown and post-lockdown timepoint compared to the pre-lockdown period (aOR = 2.96, 95% CI: 1.95–4.48) and aOR = 3.89, 95% CI: 2.65–5.72) respectively (Table 2).

The odds of immediate neonatal death was significantly higher in younger mothers aged <20 years (aOR 1.37, 95%CI: 1.06–1.77) and mothers aged 20–24 years compared to those aged 25–34 years. Immediate neonatal death was lower but not statistically significant among those aged 35+ years compared to those aged 25-34years. Women who delivered infants at <28 weeks of gestation (aOR 14.91, 10.19–21.81) or between 28–32 weeks of gestation (aOR 6.18, 4.88–7.83) compared to those who delivered infants at >32 weeks of gestations were also more likely to deliver infants who die immediately after birth (Table 3).

Additionally, the odds of stillbirths were higher in mothers over the age of 35 years (aOR = 1.25, 95% CI: 1.08–1.44) and in mothers who with >5 previous preganncies (AOR = 1.54, 95% CI: 1,22–1.94) compared to mothers aged 25–34 years and those who were Primigravida respectively. The odds of stillbirth were also higher in those born at lower

**Table 2. Maternal and infant outcomes comparing pre-lockdown, lockdown and post-lockdown time periods.**

| Characteristics | a) Pre-Lockdown Vs Lockdown | | b) Pre-Lock Vs Post lock down | |
| --- | --- | --- | --- | --- |
| | cOR* (95% CI) | aOR** (95% CI) | cOR (95% CI) | aOR (95% CI) |
| **Birth Weight of Baby (Kgs)** | | | | |
| <1.5 | 1.00 (0.84–1.20) | 0.85 (0.67–1.07) | 1.04 (0.89–1.22) | 0.89 (0.72–1.11) |
| 1.5–2.4 | 1.01 (0.91–1.12) | 0.96 (0.85–1.08) | 1.01 (0.92–1.11) | 0.95 (0.86–1.05) |
| 2.5+ | 1.00 | 1.00 | 1.00 | 1.00 |
| **Apgar Score (At 1 minute)** | | | | |
| 0 | 1.10 (0.92–1.33) | 1.05 (0.88–1.27) | 1.17 (1.00–1.38) | 1.13 (0.96–1.32) |
| 1–6 | 0.98 (0.88–1.08) | 0.96 (0.86–1.07) | 1.09 (0.97–1.23) | 1.07 (0.94–1.22) |
| 7+ | 1.00 | 1.00 | 1.00 | 1.00 |
| **Apgar Score (at 5 Minutes)** | | | | |
| 0 | 1.13 (0.95–1.35) | 1.09 (0.92–1.29) | 1.19 (1.02–1.39) | 1.15 (0.98–1.34) |
| 1–6 | 1.06 (0.90–1.25) | 1.05 (0.76–1.24) | 1.24 (1.07–1.44) | 1.22 (1.05–1.42) |
| 7+ | 1.00 | 1.00 | 1.00 | 1.00 |
| **Delivery Outcome** | | | | |
| Pre-term | 1.34 (1.20–1.49) | 1.35 (1.18–1.56) | 1.52 (1.37–1.68) | 1.67 (1.38–2.01) |
| Term | 1.00 | 1.00 | 1.00 | 1.00 |
| **Condition of baby at discharge** | | | | |
| Live Birth | 1.00 | 1.00 | 1.00 | 1.00 |
| Still Birth | 1.05 (0.90–1.23) | 1.01 (0.87–1.18) | 1.06 (0.92–1.23) | 1.03 (0.90–1.19) |
| Immediate neonatal death | 3.04 (2.02–4.57) | 2.96 (1.95–4.48) | 4.02 (2.75–5.88) | 3.89 (2.65–5.72) |
| **Obstetric Diagnosis** | | | | |
| Born Before Arrival | 1.10 (0.97–1.25) | 1.09 (0.95–1.26) | 0.42 (0.36–0.49) | 0.38 (0.31–0.45) |
| Cesarean Section | 1.12 (1.03–1.21) | 1.12 (1.02–1.21) | 1.21 (1.10–1.33) | 1.22 (1.11–1.34) |
| SVD | 1.00 | 1.00 | 1.00 | 1.00 |
| **Obstetric Complication** | | | | |
| No | 1.00 | 1.00 | 1.00 | 1.00 |
| Yes | 2.63 (2.39–2.90) | 2.62 (2.38–2.88) | 2.78 (2.54–3.05) | 2.77 (2.53–3.03) |

*Crude Odds Ratio

** Adjusted Odds Ratio

a) Pre-Lockdown Vs Lockdown is the comparison between pre-lockdown and lockdown time points;

b) Pre-Lock Vs Post lock down is the comparison between pre-lockdown and post lock lockdown time point; cOR-crude Odds Ratio; aOR- adjusted Odds Ratios for the outcomes, controlling for Mothers' Age, Gravidity, Parity and gestational age at admission.

gestational age (i.e. aOR = 6.05, 95% CI: 4.53–8.08) and aOR = 4.12, 95% CI: 3.57–4.74) for <28 weeks and 28–32 weeks respectively compared those born at gestational age >32 weeks.

**Obstetric diagnosis.** The odds of caesarian section was significantly higher during and after the lock-down compared to the time period pre-lockdown (aOR = 1.12, 95% CI: 1.02–1.21) and aOR = 1.22, 95% CI: 1.11–1.34) respectively), while the odds of infants being born before arrival to the health facility post the lockdown period decreased compared to the period pre-lockdown (aOR = 0.38, 95% CI: 0.31–0.45) (Table 2).

The odds of caesarian section was significantly lower in younger mothers (age<20 years, aOR = 0.66, 95% CI: 0.61–0.71) and those aged 20–24 years aOR = 0.86, 95% CI: 0.82–0.91) compared to mothers aged 25–34 years. The odds of undergoing caesarian section was also significantly lower among women who delivered at gestation age <28 weeks and 28–32 weeks compared to those who delivered at gestation age >32weeks (aOR = 0.42, 95% CI: 0.31–0.55 and aOR = 0.65, 95% CI 0.58–0.72 respectively) (Table 3).

**Table 3. Risk factors for selected adverse pregnancy outcomes.**

| | M1:Delivery Outcome (Preterm) aOR (95% CI) | M2: Obstetric complication (Yes) aOR (95% CI) | M3:Still Birth aOR (95% CI) | M3: Immediate Neonatal death aOR (95% CI) | M4: Caesarian Section aOR (95% CI) | M5: Born Before Arrival aOR (95% CI) |
|---|---|---|---|---|---|---|
| **Mothers Age (Years)** | | | | | | |
| <20 | 1.26 (1.14–1.40) | 0.97 (0.90–1.05) | 0.84 (0.72–0.99) | 1.37 (1.06–1.77) | 0.66 (0.61–0.71) | 2.48 (1.96–3.15) |
| 20–24 | 1.08 (1.01–1.17) | 0.99 (0.94–1.04) | 0.93 (0.83–1.05) | 1.19 (1.02–1.39) | 0.86 (0.82–0.91) | 1.80 (1.59–2.03) |
| 25–34 | 1.00 | 1.00 | 1.00 | 1.00 | 1.00 | 1.00 |
| 35+ | 1.06 (0.93–1.20) | 1.06 (0.98–1.15) | 1.25 (1.08–1.44) | 0.95 (0.75–1.20) | 1.05 (0.96–1.14) | 0.66 (0.55–0.79) |
| **Parity** | | | | | | |
| Nullipara | 1.00 | 1.00 | 1.00 | 1.00 | 1.00 | 1.00 |
| Primipara | 0.97 (0.84–1.13) | 0.94 (0.85–1.05) | 0.78 (0.63–0.95) | 1.05 (0.76–1.46) | 0.93 (0.84–1.02) | 17.8 (13.83–22.92) |
| Multipara | 1.05 (0.89–1.24) | 0.93 (0.82–1.04) | 0.83 (0.67–1.03) | 1.14 (0.77–1.69) | 0.92 (0.83–1.01) | 26.6 (19.39–36.37) |
| **Gravidity** | | | | | | |
| Primigravida | 1.00 | 1.00 | 1.00 | 1.00 | 1.00 | 1.00 |
| Multigravida-Low | 1.08 (0.93–1.25) | 1.01 (0.90–1.13) | 1.30 (1.07–1.58) | 1.22 (0.85–1.78) | 1.12 (1.02–1.23) | 0.19 (0.12–0.28) |
| Multigravida-High | 1.27 (1.05–1.55) | 1.02 (0.89–1.18) | 1.54 (1.22–1.94) | 1.38 (0.91–2.10) | 0.94 (0.84–1.05) | 0.16 (0.09–0.30) |
| **Gestational age (Weeks)** | | | | | | |
| Pre-term (<28 weeks) | | 1.42 (1.19–1.69) | 6.05 (4.53–8.08) | 14.91 (10.19–21.81) | 0.42 (0.31–0.55) | 11.08 (3.60–34.03) |
| Pre-term (28–32 weeks) | | 1.38 (1.26–1.51) | 4.12 (3.57–4.74) | 6.18 (4.88–7.83) | 0.65 (0.58–0.72) | 4.56 (2.96–7.04) |
| Term (>32 weeks) | | 1.00 | 1.00 | 1.00 | 1.00 | 1.00 |
| **Time-period** | | | | | | |
| Pre-Lockdown | 1.00 | 1.00 | 1.00 | 1.00 | 1.00 | 1.00 |
| Lock Down (period) | 1.35 (1.18–1.56) | 2.62 (2.38–2.88) | 1.01 (0.87–1.18) | 2.93 (1.93–4.46) | 1.11 (1.02–1.21) | 1.08 (0.94–1.24) |
| Post Lock down | 1.66 (1.38–2.00) | 2.77 (2.53–3.03) | 1.03 (0.89–1.18) | 3.93 (2.66–2.80) | 1.22 (1.11–1.34) | 0.35 (0.29–0.42) |

aOR- adjusted Odds Ratio; M1, M2 and M5- Binary logistic regression models while M3 &M4: Multinomial logistic regression models and used.

Gestational age excluded from M1 because they were highly correlated

## Discussion

Our study shows that it is feasible to use EHR to identify and monitor at risk subpopulation groups accessing health services during times of extreme service pressure such as outbreaks and pandemics.

Infants born preterm have higher morbidity and mortality than those born at term, with preterm birth being a leading cause of infant mortality globally [10]. Changes in preterm birth rates were reported following lockdown measures, with estimates fluctuating between -90% to +30% change in risk [11]. Reductions were described generally in high-income settings [12–15], with the inverse being reported in low some low-income settings [11, 16]. Reports of rates

of stillbirth were also mixed [11, 17]. Our study shows an increased risk of preterm birth in lockdown and post-lockdown periods in our setting and adds to the global body of evidence, though we acknowledge that missing data may limit interpretability.

We also report an increase in maternal complications and immediate neonatal death in lockdown and post-lockdown time periods. A systematic review undertaken in 2021 highlighted worsening maternal and neonatal outcomes during the pandemic, with inequity noted between high and low income settings [17]. Previous research from Uganda similarly showed an increase rate of maternal complications in the first wave of the pandemic [1], and this study confirms these findings in subsequent waves. Data from Zimbabwe showed a more resilient maternity services [18] suggesting that health care services and policy makers in low-resource settings can implement restrictive public health measures in a way which does not inadvertently harm pregnant women and children. Dialogue between institutions and agencies and across countries should be undertaken in parallel to interventions to highlight areas of good practice and share knowledge.

The direct effects of SARS-CoV-2 should also be considered. A multinational cohort study of 2,130 pregnant women across 18 countries, including Ghana and Nigeria, suggested that women with symptomatic COVID-19 during pregnancy had an increased risk of morbidity and mortality compared to non-infected pregnant women, including maternal mortality, preterm birth and pre-eclampsia [19], though other evidence from Africa has been mixed [20]. In Uganda, the Ministry of Health did not utilise KNRH as a COVID-19 treatment facility, and pregnant women who were suspected to be SARS-CoV-2 positive were moved to another hospital for treatment. It is unlikely, therefore, that the increased rate of prematurity and neonatal mortality in the lock-down period can be attributed solely to symptomatic or severe maternal infection with SARS-CoV-2. It is possible that delays in seeking antenatal and delivery care, worsened by periods of significant restrictions in public transport and a fear of acquiring COVID-19 in hospital, led to these outcomes.

Using our data, we were able to identify specific at-risk groups within the pregnant population such as younger women and those with multiple pregnancies or those admitted for labour at gestational age<32 weeks that are especially vulnerable to health services disruption [21].

Using EHR, we were able to demonstrate that, in Uganda, these changes were sustained post the easing of lock down restrictions and may suggest that the socioeconomic impact of COVID-19 had longer term indirect adverse health outcomes after the pandemic eased. Studies from Uganda have described increased food insecurity secondary to the pandemic [22]. Women, who have higher rates of unemployment and jobs that are more vulnerable to COVID-19 restrictions, were also the first to stop working during the COVID-19 pandemic [23]. This, coupled with decreased access to antenatal care during the strictest of lockdown periods [1], may have meant that women reported late to hospital and thus had an increased risk of complications during their pregnancies. This is useful information for public health bodies, seeking to target specific groups and maintain vital services during and after lockdown restrictions. We have recently reported the effect of a more localised lockdown during the Ebola outbreak in Uganda on women's health services [24].

The results of our analysis have informed development of a risk factor model and interactive dashboard for use by MOH and KNRH management to monitor changes in service use and early indicators of health system stressors. The development of a dashboard was key to disseminating the results of our study and utilized an interactive open forum comprised of teams from MOH and KNRH with expertise in MNCH services and service delivery. Engagement meetings were important prior to the development of the dashboard as they enabled critical data elements of importance nationally to be selected from the routine service delivery database systems to ease analysis and use of the data to improve service delivery. Going forward

stakeholder preferences will be key for tracking pregnancies and ensuring that the data associated with key outcomes is robust and of high quality. Electronic health records can be utilised to rapidly synthesise data in periods of rapid policy intervention to assess indirect effects and should be considered as a key tool for health monitoring. Use of EHR data to create real-time and interactive dashboards for individual facilities can increase awareness of health managers and policy makers to the impact of policy changes and alert health care workers to vulnerable patient groups.

We are limited by the use of retrospective data from a single site. KNRH is in a busy, urban area, and patterns emerging from this site may not be reflected in other health care facilities in other parts of Uganda. The use of EHR as a data source relies on health care professionals completing data capture promptly and accurately at the time of a health care encounter. Missing data, as seen in our study, requires careful consideration when interpreting trends in health outcomes. We note missing data by lockdown period and have document missing data overall due to inability to extrapolate further. The impact of missing data on inferential power of this study given the high level of missingness in routine data should be considered as the system is rolled out. Other limitations include the unavailability of important risk factors such as preexisting diabetes mellitus, hypertension, cardiovascular diseases which are important risk factors for COVID-19 disease as well as for adverse maternal and neonatal outcomes in general. However, we minimised the impact of missing data by using MICE to impute for missing data from our large dataset.

Important areas for further research include reviewing data from wider geographical settings including rural and urban areas to assess the impact of COVID-19 interventions in these areas among Adolescent Girls and Young Women (AGYW) and extending our approach to determine the impact of COVID-19 to other health areas.

## Conclusion

The effects of the COVID-19 pandemic are wide ranging, with the disruption to antenatal and maternal health care services highlighting the susceptibility of these healthcare services to wider global events. There is an urgent need for healthcare systems to plan for future pandemics or periods of insecurity in a way which limits indirect harm to vulnerable populations. Use of EHR and easy to use dashboards may enable healthcare providers and policy makers to rapidly identify at risk populations and consider health care interventions to improve health outcomes in this and future outbreaks and pandemics.

## Supporting information

**S1 Table. Level of Government restrictions on the spread of COVID-19.**
(PDF)

**S2 Table. Missing data pattern and model used.**
(PDF)

## Acknowledgments

We thank Edinance Basirika, Emily Namara Lugoloobi for supporting implementation of UgandaEMR at Kawempe National Referral Hospital. We thank the Records team led by Edirisa Muwanga and Allan Mugabi, the administration and all other staff of Kawempe National Referral Hospital for all their support and guidance as we conducted the study. We further thank Ministry of Health Uganda leadership, the technical teams from Ministry of health and

the administration of Makerere University Johns Hopkins University Research Collaboration for all their contributions, logistical and all other support rendered.

## Author Contributions

**Conceptualization:** Lauren Hookham, Kirsty Le Doare.

**Data curation:** Joseph Ouma, Lorna Aol Akera.

**Formal analysis:** Joseph Ouma.

**Funding acquisition:** Kirsty Le Doare.

**Investigation:** Lauren Hookham.

**Methodology:** Joseph Ouma.

**Project administration:** Joseph Ouma, Kirsty Le Doare.

**Supervision:** Kirsty Le Doare.

**Visualization:** Joseph Ouma.

**Writing – original draft:** Joseph Ouma.

**Writing – review & editing:** Lauren Hookham, Lorna Aol Akera, Gordon Rukundo, Mary Kyohere, Ayoub Kakande, Racheal Nakyesige, Philippa Musoke, Kirsty Le Doare.

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
