## [Decision Letter · Decision Letter 0]

26 Jun 2023

PGPH-D-23-00246

Using routine data to understand adverse pregnancy and neonatal outcomes associated with the COVID-19 pandemic in Kampala, Uganda

Dear Dr. Ouma,

Thank you for submitting your manuscript to PLOS Global Public Health. After careful consideration, we feel that it has merit but does not fully meet PLOS Global Public Health’s publication criteria as it currently stands. Therefore, we invite you to submit a revised version of the manuscript that addresses the points raised during the review process.

Please see the comments from the reviewers below. Please pay particular attention to the concerns raised about clarifying the overall aim or aims of the manuscript - whether the focus is on the effectiveness of electronic records, or the COVID-19 pandemic, or both. Please ensure that this is clear to the reader from the outset - I leave it up to you to decide how you wish to frame the study, but please include all aims in the title, abstract and introduction.

We look forward to receiving your revised manuscript.

Kind regards,

Hanna Landenmark

Staff Editor

Journal Requirements:

1. We ask that a manuscript source file is provided at Revision. Please upload your manuscript file as a .doc, .docx, .rtf or .tex.

Additional Editor Comments (if provided):

Reviewers' comments:

Reviewer's Responses to Questions

**Comments to the Author**

1. Does this manuscript meet PLOS Global Public Health’s publication criteria? Is the manuscript technically sound, and do the data support the conclusions? The manuscript must describe methodologically and ethically rigorous research with conclusions that are appropriately drawn based on the data presented.

Reviewer #1: Yes

Reviewer #2: Partly

Reviewer #3: Partly

Reviewer #4: Partly

2. Has the statistical analysis been performed appropriately and rigorously?

Reviewer #1: Yes

Reviewer #2: Yes

Reviewer #3: Yes

Reviewer #4: I don't know

3. Have the authors made all data underlying the findings in their manuscript fully available (please refer to the Data Availability Statement at the start of the manuscript PDF file)?

Reviewer #1: Yes

Reviewer #2: No

Reviewer #3: No

Reviewer #4: Yes

4. Is the manuscript presented in an intelligible fashion and written in standard English?

Reviewer #1: Yes

Reviewer #2: Yes

Reviewer #3: Yes

Reviewer #4: Yes

5. Review Comments to the Author

Reviewer #1: The aim of the research was to demonstrate the ability or capacity of the electronic health system to generate routine data that could be used to draw relationship between adverse outcomes for mothers and neonates and the COVID 19 pandemic.

Statistical analysis was well done and the data made available.

The manuscript was easy to read and presented in a well flowing manner.

No grammatical errors were identified and the language was plain with no ambiguity.

Reviewer #2: need to review the matter based on the clear research aim, in the adscript the research aim is linked with the situation of maternal health due to covid but in the conclusion parts link with effectiveness of electronic data capturing systems.

Data are well presented but its description is too less need to explain more

Conclusion and recommendation should be linked with the finding section.

Reviewer #3: The authors conducted an original study using EHR to determine maternal and neonatal outcomes during various stages of the COVID-19 pandemic (pre-lockdown, lockdown, post lockdown).

Specific comments:

Abstract:

-Can the author clarify the primary outcome in the abstract?

- Lines 38-40, authors reported the crude ORs for pre- versus lockdown not pre- versus post-lockdown of preterm and obstetric complications (Table 2). “Comparison of pre-and post-lockdown variables showed an increased risk of preterm birth (adjusted Odds Ratio (aOR) 1.34, 95% CI 1.20-1.49); obstetric complications (aOR 2.63, 95% CI 2.39-2.90)”

-What is meant obstetric complications?

Introduction:

-Can you provide the study hypothesis of the study

Methods:

-Lines 102-103, authors stated that “mode of delivery (defined as 1-born before arrival, 2-born by caesarian 103 section, or 3-born by spontaneous vaginal delivery)” what about aisted vafginal delivery? Were they removed from the analysis?

-Line 10 what is meant by obstetric complications, clarify this outcome refers to what exactly?

-How did the author specify whether the adverse outcomes were due the lockdown or pandemic itself? Are data available on COVID-19 for the mother? Was this consider in the analysis stage?

-Regarding missingness, there is a substantial amount of missing data on preterm delivery (52%) in the Pre-Lockdown stage, imputation might be not suitable for this variable.

-The authors might consider calculating the difference between the change in each outcome from the prelockdown to postlockdown periods

Results:

-I could not read data on Figure 1 & 2, may be consider increasing the resolution of them.

-Also Table S2 was not available with the submitted materials

-Lines 195-195 “There were higher proportion of children born at gestational age <32 weeks; higher

proportion of children with lower APGAR score (<7) at 1 and 5 minutes; higher proportions

women with obstetric complications; higher preterm births during and after COVID outbreak

compared to the period before COVID outbreak, Table 1”

I disagree with these interpretations, given the number of missingness on gestational age, APGAR score, you cannot confirm that there I high proportion of these outcome before after/during lockdown.

-Lines 197-198, ‘’ during and after COVID outbreak compared to the period before COVID outbreak” do you mean during and after COVID lockdown and before the lockdown period?

-Line199, correct COVI09 to COVID19

-More than 50% of data on preterm are missing in the pre-lockdown stage, were those with missingness excluded from the analysis in Table 2?

-Line 219 “The odds of preterm birth were highest in younger mothers (age<20, aOR 1.14-1.40)

compared to mothers aged 25-34 year” you write the CI, add the aOR=1.26.

_can the authors be consistent when reporting the finding, sometimes the lockdown period refers to as before the outbreak COVID 19.

-There are discrepancies between reported data in the text and that from the Table. In lines 245-247, “The odds of undergoing caesarian section were higher in younger mothers (age<20 years, aOR 2.48, 1.96-3.15 and those aged 20-24 years aOR 1.80, 1.59-2.03) compared to mothers aged 25-34 years while it was lower among older mothers aged 35+years (aOR: 0.66, 055-0.79).” author reported data related to born before arrival as data on CS (Table 3), correct these data in the text.

Discussion:

-The findings of the current study should be compared with that of the previous studies.

-The authors should highlight the limitations in the discussion, in particular, unavailable data on important risk factors, such as hypertensive disorder during pregnancy, DM covid-19 and.

-Also, the application and direction for further studies should be discussed

Other comments:

- The authors started with reference #7 in the introduction, what about reference from 1 to 6?

-References 11 & 12 are incomplete and irrelevant to the statements.

Reviewer #4: The manuscript evaluates the pregnancy outcomes pre, during, and post COVID-19 pandemic using electronic medical records. There are some major concerns that need to be addressed prior to reconsideration for publication.

- Suggest following the STROBE guidelines for observational studies

- Objective is not outlined. It is confusing to the reader if this study's primary objective is to demonstrate the use of electronic medical records or examine the pregnancy outcomes as affected by the pandemic.

- Study design is not presented in the methods section

- The variables need to be clearly defined (i.e., how is obstetric complication defined?)

- Any abbreviation should be defined initially prior to its use (abstract)

- The p-values for table 1 should be listed

- The impact of shutdown need to be addressed in the manuscript (for instance, was there an influx of patients admitted to the ICU/deployment to critical care services that would impact the care rendered to obstetric patients)

- Without having a clear understanding of the objective/aim of the study, it was difficulty to analyze the discussion and conclusion sections

- Consistency in the use of the word "COVID-19"; some parts as COVID19 and some as COVID-19

- Typo on line 199 of the manuscript

6. PLOS authors have the option to publish the peer review history of their article (what does this mean?). If published, this will include your full peer review and any attached files.

**Do you want your identity to be public for this peer review?** For information about this choice, including consent withdrawal, please see our Privacy Policy.

Reviewer #1: No

Reviewer #2: No

Reviewer #3: No

Reviewer #4: No

---

## [Decision Letter · Decision Letter 1]

25 Aug 2023

PGPH-D-23-00246R1

Using routine data to understand adverse pregnancy and neonatal outcomes associated with the COVID-19 pandemic in Kampala, Uganda

Dear Dr. Ouma,

Thank you for submitting your manuscript to PLOS Global Public Health. After careful consideration, we feel that it has merit but does not fully meet PLOS Global Public Health’s publication criteria as it currently stands. Therefore, we invite you to submit a revised version of the manuscript that addresses the points raised during the review process.

We look forward to receiving your revised manuscript.

Kind regards,

Ting Shi

Academic Editor

Journal Requirements:

1. Please amend your online detailed Financial Disclosure statement. This is published with the article. It must therefore be completed in full sentences and contain the exact wording you wish to be published.

a) State the initials, alongside each funding source, of each author to receive each grant. For example: "This work was supported by the National Institutes of Health (####### to AM; ###### to CJ) and the National Science Foundation (###### to AM)."

2. Please ensure that the funders and grant numbers match between the Financial Disclosure field and the Funding Information tab in your submission form. Note that the funders must be provided in the same order in both places as well.

3. Please update your online Competing Interests statement. If you have no competing interests to declare, please state: “The authors have declared that no competing interests exist.”

Additional Editor Comments (if provided):

Reviewers' comments:

Reviewer's Responses to Questions

**Comments to the Author**

1. If the authors have adequately addressed your comments raised in a previous round of review and you feel that this manuscript is now acceptable for publication, you may indicate that here to bypass the “Comments to the Author” section, enter your conflict of interest statement in the “Confidential to Editor” section, and submit your "Accept" recommendation.

Reviewer #1: All comments have been addressed

Reviewer #2: (No Response)

Reviewer #3: (No Response)

2. Does this manuscript meet PLOS Global Public Health’s publication criteria? Is the manuscript technically sound, and do the data support the conclusions? The manuscript must describe methodologically and ethically rigorous research with conclusions that are appropriately drawn based on the data presented.

Reviewer #1: Yes

Reviewer #2: No

Reviewer #3: (No Response)

3. Has the statistical analysis been performed appropriately and rigorously?

Reviewer #1: Yes

Reviewer #2: No

Reviewer #3: Yes

4. Have the authors made all data underlying the findings in their manuscript fully available (please refer to the Data Availability Statement at the start of the manuscript PDF file)?

Reviewer #1: Yes

Reviewer #2: (No Response)

Reviewer #3: No

5. Is the manuscript presented in an intelligible fashion and written in standard English?

Reviewer #1: Yes

Reviewer #2: No

Reviewer #3: Yes

6. Review Comments to the Author

Reviewer #1: The manuscript is presented in a legible manner. The themes flowed easily into each other and did not need any additional explanation.

Reviewer #2: 1. Information is well presented, and the author tried to review the manuscript based on comments.

2. Stil the title and conclusion are not matched

3. Methodology and analysis describe more about the importance of the records system but not the finding and analysis.

4. The analysis table is good but does not explain it and make a conclusion.

5. What will be the recommendation of this research, and what is the way forward

6. What is explained about the dashboard, and why is it essential in the study

7. Data is analysed on the electronic record system, and how they get informed consent where recoding information was taken but not explained about the data analysis and presentation.

Reviewer #3: Dr. Ouma and colleagues present a revised version of their manuscript titled " Using routine data to understand adverse pregnancy and neonatal outcomes associated with the COVID-19 pandemic in Kampala, Uganda." The authors respond to some of the previous critiques. Key edits include the following: correcting the reported aOR in the abstract. Previous version did not include a hypothesis, the authors now included their hypotheses at the conclusion of the introduction. The authors added the limitation of not considering other comorbidities (potential confounders) on the reported association. The authors also clarify the meaning of obstetric complications

Additional comments:

• If data not available of whether a delivery was assisted or not, then I suggest to change SVD to VD as we do not know whether it is a spontaneous or not.

• Clarify any abbreviation in the Tables at the footnote, such as that in Table 2

• Line 217, remove “; most of the women were multi-gravida.)”.

• In table 1, authors should provide the frequency and percentage of missing data in the related categories, so reader can be clear about the proportion of missing data in each group.

• Lines 218-219, “There were more preterm births during and after COVID-19 lock-down period compared to the pre-lock down period; Table 1.”. I still disagree with this interpretation and I think this is misleading statement as more than half of data in the pre-lockdown are missing according to the Table in the first version of the manuscript

• Lines 349 -351 “Other limitations include the unavailability of important risk factors such as pre-existing diabetes 350 mellitus, hypertension, and cardiovascular diseases which are important risk factors for COVID-19 disease.” Correct that they are important risk factors for adverse maternal and pregnancy outcomes.

• Some references require editing.

7. PLOS authors have the option to publish the peer review history of their article (what does this mean?). If published, this will include your full peer review and any attached files.

**Do you want your identity to be public for this peer review?** For information about this choice, including consent withdrawal, please see our Privacy Policy.

Reviewer #1: No

Reviewer #2: No

Reviewer #3: No

---

## [Editor Report · Decision Letter 2]

11 Oct 2023

Using electronic medical records to understand the impact of SARS-CoV-2 lockdown measures on maternal and neonatal outcomes in Kampala, Uganda

PGPH-D-23-00246R2

Dear Dr Hookham,

We are pleased to inform you that your manuscript 'Using electronic medical records to understand the impact of SARS-CoV-2 lockdown measures on maternal and neonatal outcomes in Kampala, Uganda' has been provisionally accepted for publication in PLOS Global Public Health.

Best regards,

Ting Shi

Academic Editor